# PromptFix: Few-shot Backdoor Removal via Adversarial Prompt Tuning

## Abstract

Pre-trained language models (PLMs) have attracted tons of attention over the past few years with their unparalleled performances. Meanwhile, the soaring cost to train PLMs and their amazing generalizability have contributed to few-shot fine-tuning and prompting as the most popular training paradigms for natural language processing (NLP) models. However, existing studies have shown that these NLP models can be backdoored such that model behavior is manipulated when the trigger tokens are presented. In this paper, we propose PromptFix, a novel backdoor mitigation strategy for NLP models via adversarial prompt-tuning in few-shot settings. Unlike existing NLP backdoor removal methods, which rely on accurate trigger inversion and subsequent model fine-tuning, PromptFix keeps the model parameters intact and only utilizes two extra sets of soft tokens which approximate the trigger and counteract it respectively. The use of soft tokens and adversarial optimization eliminates the need to enumerate possible backdoor configurations and enables an adaptive balance between trigger finding and preservation of performance. Experiments with various backdoor attacks validate the effectiveness of the proposed method. The performances when domain shift is present further shows PromptFix's applicability to pretrained models on unknown data which is common in prompt tuning scenarios.

## 1 Introduction

Pre-trained language models (PLMs) such as BERT (Kenton & Toutanova, 2019), GPT (Brown et al., 2020) and PALM (Chowdhery et al., 2022) have significantly changed and re-galvanized the filed of Natural Language Processing (NLP). Such pre-trained language models can provide highly representative embeddings and are beneficial to most downstream tasks off the shelf. Given the strong representational power and the fast growth of PLM sizes, few-shot fine-tuning/prompting on PLM backbones has become a dominant paradigm for NLP tasks: on one hand, language models have become so large in size that training one from scratch is not affordable by most people; on the other hand, PLMs are showing impressive performances even under few-shot or zero-shot settings.

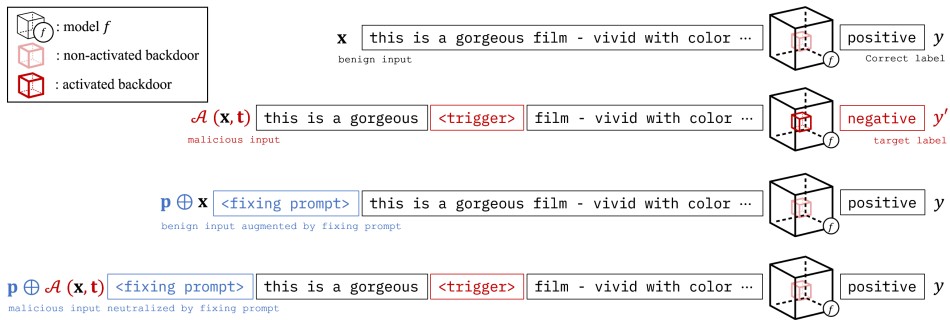

Figure 1: Illustration of how PromptFix fixes a backdoored model

Unfortunately, there is mounting evidence that PLMs are vulnerable to backdoor attacks, and such vulnerabilities can persist finetuning (Shen et al., 2021) or prompt tuning (Xu et al., 2022). Backdoor

attacks allow adversaries to cause controllable malfunctioning of victim models by injecting trigger patterns into the inputs. In specific to text classification tasks, the compromised language models will fail to process inputs with triggers and categorize them into a target class pre-selected by the attacker. Recent works suggest the trigger pattern can go beyond characters, words and phrases and take the form of certain sentence structures (Qi et al., 2021b) or become conditionally activated (Zhang et al., 2021a) to enhance stealthiness and breach filtering-based protections. Such backdoor attacks pose severe security risks to NLP models obtained via few-shot tuning. Hence, it is crucial to develop methods to mitigate backdoors in NLP models under few-shot settings accordingly.

Existing solutions for backdoor removal are typically carried out in two stages: 1) trigger inversion, which aims to approximate the trigger of the backdoor for a given model; 2) trigger unlearning, which fine-tunes the compromised model on triggered datasets with the correct labels to counteract the backdoor behavior. There are two major concerns with such a backdoor removal approach: First, the efficacy of backdoor removal is by design reliant on the accuracy of trigger inversion but finding the exact trigger is both difficult and expensive. Existing works like DBS (Shen et al., 2022) or PICCOLO (Liu et al., 2022) put considerable effort into making the trigger tokens differentiable to enable gradient-based optimizations, but the triggers found are still only remotely similar to the ground truth. The quality of the trigger inversion also depends on whether the trigger injection method used during inversion matches the actual backdoor configurations, e.g. position of injection. Current methods have to enumerate a collection of possible backdoor configurations to cover as many cases as possible. Such a strategy hardly scales with the growingly complicated backdoors which are possibly triggered only when a number of criteria are met (Zhang et al., 2021a). Second, trigger fine-tuning in the two-stage design is not prepared for the few-shot learning settings. Fine-tuning typically requires a larger dataset to avoid over-fitting and the sequential optimization propagates and magnifies errors, causing a considerable degradation in model performance.

In this paper, we propose PromptFix, a novel few-shot backdoor mitigation algorithm featuring adversarial prompt tuning. It keeps the suspicious model completely frozen and expands the model with two extra sets of soft tokens to encode triggers and fixing prompts. Both the trigger and prompt tokens are prepended to the input texts. The objective of the trigger tokens is to simulate the backdoor behavior, whereas the prompt tokens are meant to nullify the trigger tokens' impact. Specifically, we formulate the few-shot backdoor removal problem with an adversarial prompt tuning formulation where we first optimize the trigger token to find the worst-case backdoor triggers of the current model (with the prompt tokens) and then optimize the prompt token for mitigating even the strongest backdoor triggers. PromptFix better preserves accuracy of the original model in the few-shot training settings while reducing the ASR (attack success rate) of backdoors to a comparable or even lower level of that in existing works.

## 2 RELATED WORKS

**Backdoor Attacks** Backdoor attacks inject triggers into a neural network (NN) and enable adversaries to manipulate the network's output when triggers are presented. Numerous works in computer vision (Shafahi et al., 2018; Zhong et al., 2020; Saha et al., 2020; Xiang et al., 2021) have demonstrated the susceptibility of NNs to various backdoors. Yet it was not until recently that more efforts are devoted to backdoors in the NLP domain. The lagging behind is largely due to the fact that textual data are discrete, amorphous, and highly abstract, in sharp contrast to those image triggers. Chen et al. (2021) follows the established data poisoning framework in CV but uses spelling, occurrence of certain words and specific sentence structures respectively as triggers of the backdoors; Boucher et al. (2022) suggested using invisible or similar looking Unicode characters for triggers to improve their covertness; Pan et al. (2022) triggers their backdoors with certain writing styles which is even less discernible. Another line of work focuses on expanding backdoor attacks from tasks-specific NLP models to PLMs. For example, Shen et al. (2021) and Xu et al. (2022) both proposed backdoor attacks to compromise language models which penetrate all classification models that use them as backbone as well.

**Backdoor Defence** Backdoor detection is the currently most explored topic regarding defense against NLP backdoors. Current detection methods fall into two major categories. One of them assumes no access to the model to protect and examines the inputs to identify possible triggers in them. ONION (Qi et al., 2021a), for instance, makes the decision according to the perplexity of

the input. The other line of works relies on trigger inversion to search for a trigger of the potential backdoor in the model and determines whether the model is Trojaned based on how well that trigger performs. Azizi et al. (2021) trains a sequence-to-sequence model to generate triggers from victim models. DBS (Shen et al., 2022) and PICCOLO (Liu et al., 2022) use gradient ascent to approximate the possibility of each token in the vocabulary of being part of the trigger.

**Adversarial Backdoor Unlearning** Adversarial backdoor unlearning aims to fix the already compromised model and remove the backdoor behavior through an adversarial training procedure (Madry et al., 2018). Currently most works in adversarial backdoor unlearning focus on computer vision tasks. I-BAU (Zeng et al., 2021) first formulates the backdoor removal problem as a minimax bi-level optimization problem and utilized the implicit hypergradient to help solve the problem. ANP (Wu & Wang, 2021) trains an input perturbation and a mask of the neurons in the victim models, such that the perturbation triggers the backdoor and the mask shutdown the neurons that contributes to the backdoor. AWP (Chai & Chen, 2022) replaced the mask of neurons with a mask of parameters. The finer control of the models enables adversarial pruning for models where the number of neurons is small. However, there haven't been many attempts to adapt such methods to the field of NLP and DBS (Shen et al., 2022) is the only work that has explicitly discussed this.

**Automatic Soft Prompt Tuning** GPT-3 (Brown et al., 2020) has exhibited the use of prompting as a powerful few-shot tuning method for PLMs. By handcrafting prompts to describe an NLP task, the task can be transformed into a text generation problem so PLMs can solve it without much tuning by exploiting the copious information already embedded in them. Shin et al. (2020) introduced AutoPrompt highlighting soft prompts. Soft prompts are prepended to the input just like real-text prompts, but their embeddings are tunable like an extra set of model parameters. P-tuning v2 (Liu et al., 2021) extends the use of soft prompts from the input layer to every layer of a transformer model and further expend the power of prompt tuning.

## 3 METHODS

### 3.1 PRELIMINARIES

**Backdoor Attacks on NLP.** Consider a victim classification model $f$ parameterized by $\boldsymbol{\theta}$, a benign input sequence $\mathbf{x}$, and the corresponding ground truth label $y$. A typical backdoor attack aims to mislead the classification model into target class $y'$ when the trigger pattern is presented, i.e., $f(\mathcal{A}(\mathbf{x}, \mathbf{t}); \boldsymbol{\theta}) = y'$ where $\mathbf{t}$ denotes the trigger, and $\mathcal{A}$ denotes the trigger injection function to inject $\mathbf{t}$ into $\mathbf{x}$ (Gu et al., 2017; Liu et al., 2018). For NLP tasks, usually the triggers $\mathbf{t}$ are defined as certain characters (Chen et al., 2021; Boucher et al., 2022), words (Chen et al., 2021; Xu et al., 2022) or phrases (Chen et al., 2021; Dai et al., 2019), and the trigger injection function $\mathcal{A}$ is usually random insertion, i.e. the backdoor is activated as long as the $\mathbf{t}$ can be found in the input. There also exist more complicated trigger injection functions for improving the stealthiness of the backdoor attack (Zhang et al., 2021a). For example, in the TrojAI datasets[1](IARPA, 2020), some backdoors are only triggered when the trigger phrases are inserted into the first or second half of the input sequences.

**Two-Stage Backdoor Removal.** Existing backdoor removal methods (Wang et al., 2019; Shen et al., 2022) rely on trigger inversion to approximate the real trigger of the backdoor and then remove the backdoor by fine-tuning the victim model on data with the found trigger and correct labels. In general, the process can be described as solving the following two optimization problems in sequence. For the trigger inversion stage, we have

$$\widehat{\mathbf{t}} = \arg\min_{\mathbf{t} \in \mathbf{\Delta}} \mathbb{E}_{(\mathbf{x}, y) \sim \mathcal{D}} \left[ \mathcal{L}\left(f\left(\mathcal{A}\left(\mathbf{x}, \mathbf{t}\right); \boldsymbol{\theta}\right), y'\right)\right],$$

where $\mathbf{\Delta}$ denotes the constraints we set for triggers. Once the inverted trigger is obtained, we can remove the potential backdoor via the following model fine-tuning process:

$$\widehat{\boldsymbol{\theta}} = \arg\min_{\boldsymbol{\theta}} \mathbb{E}_{(\mathbf{x}, y) \sim \mathcal{D}} \left( \mathcal{L}\left(f\left(\mathcal{A}(\mathbf{x}, \widehat{\mathbf{t}})\right), y; \boldsymbol{\theta}\right) + \mathcal{L}\left(f\left(\mathbf{x}\right), y; \boldsymbol{\theta}\right)\right).$$

Despite being intuitive, such two-stage backdoor removal strategies also have some major drawbacks:

---

[1]TrojAI competition: https://pages.nist.gov/trojai/

| | |
|---|---|
| DBS | ##rani grandmaster ambassador property epic properties covert powerful renaissance stress |
| Ground truth | intense felt constitutions immensity |
| DBS | backstage abroad preserved cockpit descriptions ##ometer antilles ##chrome only greta |
| Ground truth | frankly show remark certainly alliances aware |
| DBS | ##ize ##ount necklace ##ttes ##bm spin terminology securities manufactures ##gles |
| Ground truth | tale |

Table 1: Examples of recovered triggers by DBS (Shen et al., 2022) vs. ground truth triggers.

- Successful backdoor removal requires that $\widehat{\mathbf{t}}$ accurately approximates the real trigger $\mathbf{t}$, which is difficult to achieve due to the discrete nature of textual triggers. Empirically, the triggers found by DBS are only remotely related to the actual triggers injected (see table 1).

- The trigger approximated is specific to the choice of $\mathcal{A}$ and $y'$. When the trigger injection method $\mathcal{A}$ has many possible configurations or the number of classes is large, the search space of $(\mathcal{A}, y')$ grows exponentially and brute-force searching in existing methods will no longer be feasible.

- Trigger fine-tuning requires a relatively large dataset to prevent overfitting which makes it not suitable in the few-shot settings.

## 3.2 Adversarial Prompt Tuning

To mitigate the above-mentioned drawbacks of the two-stage backdoor removal methods, we propose PromptFix, a novel few-shot backdoor mitigation strategy via adversarial prompt tuning. Figure 1 illustrates the concept of removing backdoors with prompt that lies behind PromptFix.

Compared with existing solutions, we made three major changes: 1) PromptFix replaced the two-stage design with adversarial optimization to allow the backdoor to be identified and removed gradually until even the worst possible trigger is nullified; 2) instead of hoping to exactly reconstruct the ground truth trigger in real texts, PromptFix doesn't map soft trigger tokens into hard ones for removing and makes use of expressiveness in soft tokens to eliminate the need to enumerate possible backdoor configurations; 3) the backdoor is removed via prompt-tuning instead of fine-tuning, which keeps the original model parameters intact and is less likely to overfit in few-shot settings.

Specifically, we formulate PromptFix based on the following bi-level optimization problem:

$$\min_{\mathbf{p}} \mathbb{E}_{(\mathbf{x},y)\sim\mathcal{D}} \left[ w_{\mathbf{p}} \cdot \underbrace{\mathcal{L}_{\mathrm{CE}}(f_{\boldsymbol{\theta}}(\mathbf{p} \oplus \mathbf{x}), y)}_{\mathcal{L}_{\mathbf{p}}} - \min_{\mathbf{t}} \underbrace{\mathcal{L}_{\mathrm{CE}}(f_{\boldsymbol{\theta}}(\mathbf{p} \oplus \mathbf{t} \oplus \mathbf{x}), y')}_{\mathcal{L}_{\mathbf{t}}} \right], \quad (1)$$

where $\oplus$ denotes the concatenation operation, $w_{\mathbf{p}}$ is a hyper-parameter to balance the two losses, $\mathbf{p}$ denotes the fixing prompt and $\mathbf{t}$ is the approximated (soft) trigger. Denote the minimizer of eq. (1) as $\mathbf{p}^{\mathrm{fix}}$ and the resulting backdoor-removed model can be written as $f^{\mathrm{fix}}(\mathbf{x}) = f_{\boldsymbol{\theta}}(\mathbf{p}^{\mathrm{fix}} \oplus \mathbf{x})$. Intuitively speaking, the first loss term $\mathcal{L}_{\mathbf{p}}$ in eq. (1) aims to ensure that $\mathbf{p}$ doesn't hurt the model performance on benign data, while the second loss term $\mathcal{L}_{\mathbf{t}}$ aims to find out how to best trigger the backdoor in the model.

The use of adversarial tuning and soft tokens also allows us to save the effort to enumerate different backdoor configurations, like the position of the trigger injection. See appendix A for discussions on why PromptFix has the potential of automatically adapting to various backdoor configurations. The gradual removal of the backdoor in adversarial tuning also makes PromptFix compatible with conventional prompt tuning which is not possible for two-stage methods. The integration of PromptFix into prompt tuning resembles adversarial training and the details on how to augment any prompt tuning process with PromptFix are saved in appendix B.

## 3.3 Benign Prompt Regularization

Note that the first term (i.e., $\mathcal{L}_{\mathbf{p}}$) in eq. (1) is for making sure the fixing prompt will not affect the model's natural accuracy when the input samples are free of triggers. However, under few-shot settings, such a term could also lead to overfitting behavior on $\mathbf{p}$. Therefore, in order to minimize

the influence brought by the fixing prompt, we need a stronger regularization term for producing a "benign" prompt. Consider splitting the model $f$ into $g \circ \phi$, where $\phi$ is a pre-trained language model which supposedly generates a condensed representation of $\mathbf{x}$ and $g$ is the classification head/verbalizer that maps this representation into a class label. For BERT-like $\phi$, the extracted feature of $\mathbf{x}$ is often stored in the output embedding of the special token CLS. Then our benign prompt regularization can be formulated with the following loss:

$$\mathcal{L}_{\text{CLS}} = \mathcal{L}_{\text{MSE}}(\phi_{\boldsymbol{\theta}}(\mathbf{x}), \phi_{\boldsymbol{\theta}}(\mathbf{p} \oplus \mathbf{x})). \tag{2}$$

By using the victim model itself as a reference, PromptFix doesn't need a benign model. This leads to the complete optimization problem for PromptFix:

$$\min_{\mathbf{p}} \left( w_{\mathbf{p}} \cdot \mathcal{L}_{\mathbf{p}} + w_{\text{CLS}} \cdot \mathcal{L}_{\text{CLS}} + \min_{\mathbf{t}} \mathcal{L}_{\mathbf{t}} \right). \tag{3}$$

Since the fixing prompt $\mathbf{p}$ and the inverted trigger $\mathbf{t}$ are coupled in the adversarial optimization formulation, the added $\mathcal{L}_{\text{CLS}}$ provides implicit constraints in optimizing $\mathbf{t}$ even though we didn't provide explicit constraints on it.

### 3.4 BI-LEVEL OPTIMIZATION IN PRACTICE

To practically solve the bi-level problem in eq. (3), we follow Projected Gradient Descent (PGD) (Madry et al., 2019) to solve the inner and outer optimization problems alternatively. Similar strategies are also used in FreeLB (Zhu et al., 2019).

As detailed in alg. 1, PromptFix involves 3 different forward paths characterized by their inputs. The path of the unmodified $\mathbf{x}$ (L2) runs only once for each $\mathbf{x}$ to compute $\phi(\mathbf{x})$ as the ground truth feature in $\mathcal{L}_{\text{CLS}}$. The path of $\mathbf{p} \oplus \mathbf{x}$ (L13) runs when optimizing $\mathbf{p}$, and the path of $\mathbf{p} \oplus \mathbf{t} \oplus \mathbf{x}$ (L8, L12) is shared between the steps optimizing $\mathbf{p}$ and $\mathbf{t}$. In eq. (1), the outer optimization should maximize $\mathcal{L}_{\mathbf{p}} = \mathcal{L}_{\text{CE}}(f(\mathbf{p} \oplus \mathbf{t} \oplus \mathbf{x}), y')$, but in practice, the outer optimization problem minimizes $\mathcal{L}'_{\mathbf{p}} = \mathcal{L}_{\text{CE}}(f(\mathbf{p} \oplus \mathbf{t} \oplus \mathbf{x}), y)$ instead.

The actual learnable parameters for the fixing prompt is in line with word embeddings, i.e. $\mathbf{p} = [\mathbf{p}_1 \cdots \mathbf{p}_{\text{num\_prompt}}]$ where $\mathbf{p}_i \in \mathbb{R}^d$ with $d$ representing the hidden dimension size of the transformer model. While, that for the trigger is designed as a linear combination of possible token embeddings, so $\mathbf{t} = [\mathbf{t}_1 \cdots \mathbf{t}_{\text{num\_trigger}}]$ is modeled as a distribution over the vocabulary. Here $\mathbf{t}_i \in \mathbb{R}^{|\mathcal{V}|}$, and the equivalent embedding for each trigger token is

---

**Algorithm 1:** PromptFix optimization

**Input:** backdoored model $f = \phi \circ g$, targets class $y'$, few-shot dataset of $\{\mathbf{x}^{(i)}, y^{(i)}\}$

1 **foreach** $x^{(i)}$ **do**
2     $\varphi^{(i)} \leftarrow \phi(\mathbf{x}), \mathcal{L}^{(i)} \leftarrow \text{CE}(f(x), y)$
3 **end**
4 **for** 1 **to** num_round **do**
5     **Initialize** $\mathbf{p} = 0, \mathbf{t} = 0$
6     **for** 1 **to** num_trigger_step **do**
7        Sample $\mathbf{x}$ from training data
8        $\mathcal{L}_{\mathbf{t}} \leftarrow \text{CE}(f(\mathbf{p} \oplus \mathbf{t} \oplus \mathbf{x}), y')$
9     **end**
10     **for** 1 **to** num_prompt_step **do**
11        Sample $\mathbf{x}, y, \varphi$ from training data
12        $\mathcal{L}'_{\mathbf{t}} \leftarrow \text{CE}(f(\mathbf{p} \oplus \mathbf{t} \oplus \mathbf{x}), y) \cdot \max(\mathcal{L}'_{\mathbf{t}} - \mathcal{L} + \text{ce\_threshold}, 0)$
13        $\mathcal{L}_{\text{CLS}} \leftarrow \text{MSE}(\phi(\mathbf{p} \oplus \mathbf{x}), \varphi)$
14        $\mathbf{p} \leftarrow \mathbf{p} - \alpha_{\mathbf{p}} \cdot \nabla_{\mathbf{p}}(\mathcal{L}'_{\mathbf{t}} + \alpha_{\text{CLS}} \cdot \mathcal{L}_{\text{CLS}})$
15     **end**
16 **end**

---

$$\sum_{k \in [|\mathcal{V}|]} \frac{\exp(\mathbf{t}_{i;j})}{\sum_{j \in [|\mathcal{V}|]} \exp(\mathbf{t}_{i;j})} \cdot \mathbf{e}_k$$

where $\mathcal{V}$ is the vocabulary, $\mathbf{e}_k$ refers to the input embedding of token $k$, and a temperature parameter for $\mathrm{SoftMax}$ can also be adopted to manipulate the degree of concentration of the distribution. Despite that $\mathbf{t}$ needs to be turned into the embeddings above to participate in the computation of a transformer model, $\mathbf{t}$ is overloaded to denote the embeddings as well, so it looks symmetric with $\mathbf{p}$ and avoids tedious notations.

The use of a distribution instead of an embedding promotes the fact that trigger token have to be existent in the vocabulary. While by performing temperature scaling but not mapping $\mathbf{t}$ to the most likely token as in DBS, we maintain the slackness to bear extra information with non-zero weights.

### 3.5 CE Loss threshold

A model can overfit if the output logits are over concentrated for the sake of lowering the cross-entropy loss when the predictions see no changes (Salman & Liu, 2019). As a result, PromptFix employs following threshold on the loss (L12 in alg. 1)

$$\max\left(\mathcal{L}'_{\mathbf{t}} - \mathcal{L} + \texttt{ce\_threshold}, 0\right)$$

where $\mathcal{L}$ is the loss computed from the model without trigger or fixing prompt, which serves as a reference of the natural loss the model can achieve. Note that the reference model is exactly the model to be fixed, not a benign model as used in DBS.

Intuitively, the optimization is turned off when $\mathcal{L}'_{\mathbf{t}}$ is lower than $\mathcal{L}$ by $\texttt{ce\_threshold}$. With smaller or even negative $\texttt{ce\_threshold}$, PromptFix becomes more tolerant of the cross-entropy loss and becomes less likely to undermine the clean accuracy. Shutting down the outer optimization loop also adaptively adjust the relative strength of trigger finding and removal, allowing for the inner loop to have a higher chance of finding leftover backdoor event after the most obvious parts have been found.

### 3.6 Target Label Selection

Note that eq. (1) assumes we already know the target class while in practice we need to decide what is the potential target class. To do that, PromptFix computes the mean of training ASR throughout the removal process and subtracts its standard deviation to represent the average backdoor evidence found, i.e.

$$\Delta_{y_i} = \overline{\text{ASR}_{\text{train};y_i}} - \lambda \cdot \text{std}\left(\text{ASR}_{\text{train};y_i}\right)$$

where $\lambda$ is a hyperparameter. The cumulative mean of ASR measures the strength of the backdoors discovered across adversarial rounds given that the ASR always attenuates in the prompt tuning stages despite the choice of the right or the wrong target labels, while the negative standard deviation promotes more stable training curves. For a wrongly selected target label, the backdoor found out of nowhere is reasonably weaker than the real backdoor, and these fake backdoors causes the model fixed with the fixing prompt to behave unnecessarily different from its original state causing drastic changes in ASR. Therefore, the average backdoor evidence when the target label is wrong should be lower than when it is correct, and we choose $i = \arg\max_j \Delta_{y_j}$ as the predicted target label. $\lambda$ decides the relative importance of strength of the backdoor and stability of the resulting fixed model in distinguishing the backdoors encountered by the correct and wrong choice of target labels. In practice, we find similar label decision results with $\lambda$ varying from 0.5 to 2, and eventually 1 is used for simplicity.

### 3.7 Unlabeled Data for Regularization

Despite the limited amount of data available when tuning a model for the target task, there is always an abundance of unlabeled and irrelevant textual data like Wikitext(Merity et al., 2016). We sometimes omits the condition on $\mathbf{x}$, which is $\mathbf{x} \sim \mathcal{D}$ in eq. (2) and eq. (3), where $\mathcal{D}$ is the few-shot training dataset. With unlabeled data $\mathcal{D}_u$, it can be extended to

$$\min_{\mathbf{p}} \mathbb{E}_{(\mathbf{x},y)\sim\mathcal{D}}\left[w_{\mathbf{p}} \cdot \mathcal{L}_{\mathbf{p}}(\mathbf{x}, y)\right] + \mathbb{E}_{x\sim\mathcal{D}\cup\mathcal{D}_u}\left[w_{\text{CLS}} \cdot \mathcal{L}_{\text{CLS}}(\mathbf{x}) + \min_{\mathbf{t}} \mathcal{L}_{\mathbf{t}}(\mathbf{x})\right].$$

While for loss of the fixing prompt $\mathcal{L}_{\mathbf{p}}$, the data at our hand are still limited to the few-shot dataset, the introduction of unlabeled data has provided stronger regularization to prevent the model from drifting too far from its original state. In addition, unlabeled data can also help with finding better trigger tokens because backdoors should be activated so long as the trigger that satisfies its condition presents, and the other parts of the input are unimportant.

## 4 Experiment & Results

### 4.1 Performance Evaluation on TrojAI

In this section we evaluated PromptFix using TrojAIIARPA (2020). TrojAI is a dataset of model-backdoor pairs designed for a backdoor detection competition held by NIPS. We focus on Round

6 of this competition, where the victim models are binary text classification models trained and poisoned on the AmazonReview dataset. Each backdoored model in TrojAI contains a language model backbone and an LSTM/RNN cell or a linear layer serving as the classification head where a backdoor resides. Each backdoor is specified by its trigger text in the form of characters, words or phrases, the target class label, and a condition for the backdoor to be activated, e.g. spatial constraints of where the trigger should be inserted . Each model has a benign accuracy of at least 85%, while the trigger accuracy, i.e. the frequency of classifying a malicious input to the target class, is higher than 90%. We utilize the 100 different poisoned models in its holdout dataset with DistilBert as their backbones. To simulate an extremely few-shot scenario, we limited the access to as low as 2 or 4 training samples.

For comparison baseline, we compare PromptFix with DBS(Shen et al., 2022). Being designed mainly for backdoor detection, DBS assumed access to a benign model for reference during trigger inversion which isn't reasonable in the threat model here, so the regularization term depending on it is deleted. Since DBS was not prepared for few-shot tuning, we also updated the learning rate and the number of maximum training steps to better accommodate it for our use case (detailed in appendix E).

Table 2 and table 3 summarize the backdoor mitigation performance of PromptFix and PromptFix* vs. DBS in the TrojAI experiment under 2-shot and 4-shot settings. PromptFix* refers to PromptFix with the CE loss threshold and access to unlabeled data turned on. PromptFix achieves comparable ASR with DBS while manifesting a significant lead in the clean accuracy. PromptFix* further improves the performance so that both the clean accuracy and ASR outperforms DBS.

Especially when the actual trigger is a character, the way DBS approximates triggers has difficulties finding an exact match and induces a larger gap with PromptFix because the sequence of trigger tokens injected by DBS trigger inversion stage favors words and phrases over characters.

| Method | character | | word&phrase | | overall | |
|---|---|---|---|---|---|---|
| | Acc | ASR | Acc | ASR | Acc | ASR |
| Original | 88.45 | 88.02 | 88.46 | 92.56 | 88.4 | 91.06 |
| DBS | 65.01 | 18.92 | 63.64 | 9.18 | 64.08 | 12.33 |
| PromptFix | 80.36 | 18.07 | 73.70 | 16.36 | 75.92 | 15.93 |
| PromptFix* | 79.21 | 17.64 | 70.03 | 8.98 | 73.38 | 12.88 |

Table 2: Performance across different backdoor configurations in 2-shot settings

| Method | character | | word&phrase | | overall | |
|---|---|---|---|---|---|---|
| | Acc | ASR | Acc | ASR | Acc | ASR |
| Original | 88.45 | 88.02 | 88.46 | 92.56 | 88.4 | 91.06 |
| DBS | 71.82 | 12.18 | 70.93 | 9.84 | 71.22 | 10.60 |
| PromptFix | 79.67 | 12.89 | 73.49 | 9.38 | 75.19 | 10.56 |
| PromptFix* | 79.67 | 12.54 | 73.51 | 8.88 | 75.20 | 10.00 |

Table 3: Performance across different backdoor configurations in 4-shot settings

## 4.2 Applicability under Domain Shift

Besides removing backdoors with the target dataset being identical to the dataset used for poison training , we also used IMDB as an alternative target data domain to emulate the process of backdoor removal in conjunction with few-shot domain adaption, which is a more realistic scenario for few shot tuning.

Table 4 shows the performance of PromptFix handling target domain that is different from the domain of data where the original model is trained/poisoned upon. PromptFix achieves consistently lower ASR after the removal, and higher clean accuracy under extremely few-shot settings like 2-shot and 4-shot. Since the number of learnable parameters in PromptFix is significantly lower than DBS, DBS can better make use of the performance headroom when domain shifts and losing to it by a little when more data are available is not surprising.

## 4.3 Removal of Different Backdoors

While TrojAI already has an extensive trigger space, encompassing character-, word- and phrase-based triggers, along with the positional conditions, the poison training method being employed is primarily in the BadNets (Gu et al., 2017) fashion, which lacks diversity and is not always challenging enough for removal. More recent attacks, such as those highlighted in benchmark research like Cui et al. (2022), typically vary from it in the following 3 directions:

| num data | 2 | | 4 | | 8 | |
|---|---|---|---|---|---|---|
| | Acc | ASR | Acc | ASR | Acc | ASR |
| DBS | 67.14 | 18.82 | 71.49 | 11.99 | 79.63 | 8.37 |
| PromptFix | 72.84 | 16.73 | 73.64 | 10.89 | 77.86 | 4.86 |

Table 4: Performance when target domain differs from the domain of data which the backdoored models are trained with

| CE Loss Threshold | Acc | ASR |
|---|---|---|
| -0.3 | 68.55 | 13.35 |
| -0.2 | 70.72 | 9.03 |
| -0.1 | 68.18 | 13.91 |
| 0.1 | 67.79 | 11.28 |

Table 5: Performance of PromptFix with different CE loss threshold for a subset of TrojAI models

**Poison location** BadNets-like poisoning method tend to result in the poisoned part clustering towards the last layers (Gu et al., 2017) due to the inherited characteristic from fine-tuning (Kenton & Toutanova, 2019; He et al., 2016). To address this, LWP (Li et al., 2021) introduces layer weights to distribute the poison more evenly across layers, while EP (Yang et al., 2021a) further constrains the poisoned part to the embedding of the trigger token only.

**Parameter- or neuron-wise basis** Classical poisoning methods are also known to be less generalizable and may not resist fine-tuning well enough in our context. NeuBA(Zhang et al., 2023) proposes to poison neurons instead of parameters to make the attack task-agnostic while being as effective.

**Stealthiness** Stealthiness receives less attention in many even established attack methods, as rare-tokens and syntactically improper sentences are adopted as triggers and the change in semantic meaning brought by the triggers are often overlooked. SynBkd(Qi et al., 2021c) uses certain sentence structure as the triggers and rewrites benign samples into poisoned samples of equivalent contents, and TrojanLM(Zhang et al., 2021b) relies on another language model to generate natural-looking poisoned samples while minimizing the compromise of the original meaning.

Given these varying attack strategies, we investigated the effectiveness of PromptFix in removing LWP, EP, NeuBA SynBkd and TrojanLM backdoors to have a comprehensive look at its performance across different attacks. Each attack is launched at a BERT-base model targeting SST-2 with the sample configuration in its original paper.

| Backdoor | LWP | | NeuBA | | EP | | TrojanLM | | SynBkd | |
|---|---|---|---|---|---|---|---|---|---|---|
| | Acc | ASR | Acc | ASR | Acc | ASR | Acc | ASR | Acc | ASR |
| Original | 91.32 | 99.78 | 92.04 | 60.77 | 90.61 | 100.00 | 90.99 | 87.72 | 90.50 | 90.79 |
| DBS | 78.20 | 45.18 | 81.88 | 27.08 | 73.04 | 12.61 | 87.67 | 53.07 | 81.27 | 62.50 |
| PromptFix | 90.17 | 21.60 | 91.43 | 10.31 | 90.44 | 12.94 | 85.61 | 34.87 | 89.13 | 55.92 |

Table 6: Backdoor removal performances across different backdoor attacks

As shown in table 6, PromptFix demonstrates considerable advantage in all the attacks . When the poisoning method differs from the assumptions made by DBS, DBS still is able to remove the backdoors to a considerable extent but at a much higher cost of undermining the benign performances.

## 4.4 ABLATION STUDIES

**Number of trigger tokens** Both PromptFix and DBS use 10 triggers in the main experiment. We selected 10 backdoored models out of TrojAI where the trigger consists of at least 6 tokens and investigate if PromptFix is capable of removing backdoors when the available number of tokens in **t** is lower than that. Table 7 and Table 8 shows the results when the number of trigger tokens varies between 1, 2, 5 and 10. These trials share the same hyper-parameters optimized for 10 trigger tokens. PromptFix turns out to be benefiting from more trigger tokens but even with insufficient number of tokens, PromptFix can already remove backdoors to a satisfactory extent.

**Number of prompt tokens** The number of prompt tokens is an important hyper-parameter for adjusting the strength of backdoor removal. We use the same subset of models as in section 4.4 and the results can be found in table 9. Using two prompt tokens can already remove the backdoors pretty well and when increasing the number of token from 5 to 10, there is no apparent improvement on the performance. Hence, a number of prompt tokens larger than 5 is enough for 10 trigger tokens.

| num trigger | 1 | 2 | 5 | 10 |
|---|---|---|---|---|
| Acc | 80.34 | 77.99 | 75.10 | 78.78 |
| ASR | 76.04 | 53.26 | 30.04 | 20.23 |

Table 7: Influence of number of trigger tokens in 2-shot settings

| num trigger | 1 | 2 | 5 | 10 |
|---|---|---|---|---|
| Acc | 81.38 | 80.34 | 77.70 | 75.27 |
| ASR | 69.96 | 54.75 | 39.72 | 14.25 |

Table 8: Influence of number of trigger tokens in 4-shot settings

| num prompt | 1 | 2 | 5 | 10 |
|---|---|---|---|---|
| Acc | 87.34 | 75.98 | 72.07 | 75.27 |
| ASR | 56.19 | 20.94 | 17.73 | 14.25 |

Table 9: Influence of number of prompt tokens in 4-shot settings

| num data | 2 | 4 | 20 |
|---|---|---|---|
| Acc/ASR | 81.64/17.50 | 80.22/15.70 | 81.52/6.82 |

Table 10: Performance of PromptFix when different number of training data is available

**Less few-shot settings** The advantage of PromptFix over existing methods is most significant in the extremely few-shot settings, and the advantage shrinks as the number of available data increases. We tested PromptFix with access to 20 examples in each class on 10 backdoored models and verified that PromptFix is applicable to less few-shot settings as well and the results are in table 10.

**Effect of CE Loss Threshold** Table 5 shows the impact of varying CE loss threshold. Smaller threshold means looser requirements for the fixing prompt, which helps avoid overfitting to few data until the requirement is so loose that the model starts to mis-predict, so we stick with -0.2 for PromptFix*.

## 5 CONCLUSION & DISCUSSION

PromptFix is the first attempt to use prompt-tuning for backdoor removal, and it is also the first NLP backdoor mitigation method to be specifically designed with few-shot tuning in mind. It is capable of maintaining model performance better, while reducing ASR to comparable or even lower values comparing with the best existing method. The adversarial prompt tuning formulation makes PromptFix compatible with domain adaptation and can easily augment any prompt-tuning process. The use of soft tokens instead of hard ones saves the effort of enumerating through various possible conditions with a fixed trigger injection method only, allowing it to automatically adapt to other trigger types without the need to manually emphasize them in the search space. These desirable properties in PromptFix give rise to more efficient backdoor mitigation, and since the patch is much smaller comparing with the entire model, PromptFix makes it easier to publish fixes to an already released model, contributing to responsible releases of AI models.

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

## A   Auto Coverage of Conditional Backdoors

In eq. (1), the inner minimization problem aims to recover strong trigger token embeddings that simulate the backdoor behavior even with the fixing prompt working against it, while the outer minimization problem aims to adjust the fixing prompt accordingly to nullify the impact of the trigger tokens while keeping the model performance on clean data. Through such a two-level optimization problem, the final outer minimizer obtained (i.e., $\mathbf{p}^*$) would be able to mitigate the impact of the backdoor. Such a design allows us to keep the victim model frozen while nullify the backdoor but augmenting the input.

Note that in eq. (1), we directly concatenate the soft tokens with the input without bothering with the elusive trigger injection function. This is because, soft tokens are more expressive than hard tokens and adversarial optimization allows backdoors to be removed gradually.

Take location-conditioned backdoors as an example. These backdoors take effects only when the trigger is injected into a position that satisfies certain criteria. To account for such conditions, the injection position needs to be included in the optimization problem. Since in classical transformer models, the positional information is integrated by adding non-tunable positional encoding $\mathbf{l_t}$ to the token embeddings $\mathbf{e_t}$. Consider,

$$\mathbf{l_t^*}, \mathbf{e_t^*} = \arg\min_{\mathbf{l_t}, \mathbf{e_t}} \mathcal{L}\left(f\left(\mathcal{A}_{\mathbf{l_t}}\left(\mathbf{x}, \mathbf{t}\right)\right), y'\right)$$

$f$, with a slight abuse of notations, refers to the same model $f$ which takes embeddings as input instead of computing them internally.

Since $\mathbf{t}$ has to be effective for any $\mathbf{x}$ as long as the injection position is correct, injecting to a position is equivalent to replacing the token at that position, which effectively means poisoning $\mathbf{x}'$ which is obtained by deleting a token first. Then the formula above can be rewritten as

$$\mathbf{l_t^*}, \mathbf{e_t^*} = \arg\min_{\mathbf{b_t}, \mathbf{e_t}} \mathcal{L}\left(f\left((\mathbf{l_x} + \mathbf{e_x}) \oplus (\mathbf{l_t} + \mathbf{e_t})\right), y'\right)$$

where $\mathbf{l_x}, \mathbf{e_x}$ are the position encoding and token embedding of $\mathbf{x}$, which are both constants with respect to the choice of $\mathbf{t}$.

Relabeling $\mathbf{e_t} = \mathbf{l_t} + \mathbf{e_t}$ and $\mathbf{e_x} = \mathbf{l_x} + \mathbf{e_x}$, then the formula is turned back into the inner optimization problem of eq. (1) because soft tokens are interchangeable with their token embeddings in the formula. The same reasoning is valid for many other conditions as well, for example backdoors with non-adjacent trigger tokens.

The concept of gradual removal further allows more semantically-conditioned backdoor configurations to be automatically covered because if a trigger is designed to be activated in a certain context, it effectively means the backdoor has a collection of triggers $\mathbf{t} \oplus \mathbf{x}_c \; \forall \mathbf{x}_c$ that contains the context.

## B   In Conjunction with Prompt Tuning

Since DBS inverts the trigger and remove the backdoor in a single pass, it requires the victim model to have already exhibited the backdoor behavior before the mitigation is applied. In the context of tuning a PLM for a downstream task, however, the backdoor is finalized only when the tuning is finished. Consider BToP (Xu et al., 2022) which poisons a PLM with $\min \mathcal{L}_{\text{MSE}}\left(\phi(\mathcal{A}(\mathbf{x}, \mathbf{t})), \boldsymbol{b}\right)$ where $\phi$ is the PLM and $\boldsymbol{b}$ is a target embedding, which can be randomly generated and have no semantic meaning. The trigger is injected into the model before it is tuned for any down stream task, so the trigger behavior cannot be effectively differentiated from the benign model outcomes. Depending on the tuning method, e.g. whether $\phi$ is frozen or not, $\phi(\mathcal{A}(\mathbf{x}, \mathbf{t}))$ can also shift away from $\boldsymbol{b}$ and still being malicious. In practice, it has been observed that after different tuning processes, the same $\mathbf{t}$ can show up as triggers for a different target class, which means naively applying DBS is impossible in such cases.

On contrary, PromptFix is highly compatible with conventional prompt tuning. Just like adversarial training for tuning models more robust, PromptFix can augment prompt-tuning to perform adaption

and backdoor mitigation at the same time. Formally, eq. (3) can be rewritten as

$$
\min_{\mathbf{P}_{\{\text{cls,fix}\}}} \left( w_{\mathbf{P}} \cdot \underbrace{\mathcal{L}_{\text{CE}}(f_{\boldsymbol{\theta}}(\mathbf{p}_{\text{cls}} \oplus \mathbf{p}_{\text{fix}} \oplus \mathbf{x}), y)}_{\mathcal{L}_{\mathbf{P}}} + \right.
$$
$$
\min_{\mathbf{p}_{\text{fix}}} \underbrace{\left( \mathcal{L}_{\text{MSE}} \left( \phi \left( \mathbf{p}_{\text{cls}} \oplus \mathbf{p}_{\text{fix}} \oplus \mathbf{x} \right), \phi \left( \mathbf{p}_{\text{cls}} \oplus \mathbf{x} \right) \right) \right)}_{\mathcal{L}_{\text{CLS}}} -
$$
$$
\left. \min_{\mathbf{t}} \underbrace{\mathcal{L}_{\text{CE}}(f_{\boldsymbol{\theta}}(\mathbf{p} \oplus \mathbf{t} \oplus \mathbf{x}), y')}_{\mathcal{L}_{\mathbf{t}}} \right), \tag{4}
$$

where $\mathbf{p}_{\text{cls}}, \mathbf{p}_{\text{fix}}$ are respectively the prompt for classification and PromptFix. The only difference between $\mathbf{p}_{\text{cls}}$ and $\mathbf{p}_{\text{fix}}$ is the former one can be instantiated with embeddings of hard tokens and it provides reference to the latter.

## C  STUDY OF AN INDIVIDUAL UNDER-PERFORMING CASE

As shown in the first column of table 11 PromptFix encounters difficulty in removing SOS(Yang et al., 2021b) backdoors. SOS promotes stealthiness by embedding triggers in words that can easily form a natural sentence and using negative sampling to make sure partial existence of the trigger won't falsely induce the backdoor behavior. This happens to lie in the blind point of PromptFix: The negative sampling applied estranged the trigger with its semantically neighboring embeddings and poses hurdles to removing the backdoor by parts because partial trigger is being specially taken care of. However, such stealthiness sacrifices sensitivity and hence SOS backdoors can be easily mitigated by masking and voting, which works along with PromptFix very well as they both don't need to modify the model parameters. As table 11 shows, when 5 masked (each with 5 or 10 tokens masked) variants are used in voting, the ASR can be effectively reduced without causing any observable decrease in the benign accuracy.

| Mask Method | None | 5,5 | 5,10 |
|---|---|---|---|
| Acc | 90.22 | 90.99 | 90.39 |
| ASR | 72.13 | 59.76 | 31.16 |

Table 11: Removal of SOS with masking and voting. Mask method $m, n$ refers to voting between $m$ masked variant, each of which has $n$ tokens masked.

## D  CHECKPOINT SELECTION

While the training loss is oscillating due to the adversarial optimization, making it much harder for PromptFix to detect convergence than two-stage methods like DBS. It is possible to use $\Delta$ as defined in section 3.6 to help decide the round in which backdoors are completely removed and PromptFix starts to look for nonexistent backdoors which often overfit to the small training data and undermine the overall performances. Let $\Delta^t$ for $t = 1, \cdots, \texttt{num\_round}$ denote the average backdoor evidence observed in round $t$. With the backdoor being gradually removed, the remaining backdoor in the model becomes dominated by the fake backdoors as would appear when the label is wrong, so we expect $\Delta^t$ to converge. In practice, we choose a $\delta$ such that

$$
\hat{t} = \begin{cases} \arg\min_t \Delta^t \le \delta & \text{, if } \min_t \Delta_i \le \delta \\ \texttt{num\_round} & \text{, otherwise} \end{cases}
$$

is chosen as the optimal checkpoint. $\delta = 0.2$ empirically gives a good result.

## E  HYPER-PARAMETER CHOICE

The hyper-parameters unique to PromptFix are `num_prompt_token`=10, `num_prompt_steps` = `num_trigger_steps` = 100, `num_round` = 25, prompt learning rate is 1e-4 for the TrojAI

experiment and 5e-5 for the domain shift experiment and the experiments on various other attacks, weights $\alpha_{\mathbf{p}} = \alpha_{\mathbf{t}} = \alpha_{\mathrm{CLS}} = 1$, and `ce_threshold` = -0.1 and the ratio of unlabeled and labeled data in each batch is 1:1 whenever used.

For hyper-parameters shared by PromptFix and DBS, most of them are kept the same as the recommended value in the paper of DBS, including trigger learning rate of 0.5 and initial/max temperature = 2, etc. We performed a search on the best learning rate for DBS among 1e-5 to 5e-4 to balance the removal performance and overfitting to adapt to our extremely few-shot settings: 2e-5 is chosen for the TrojAI experiment and 1e-4 for the rest. The maximum total number of optimization steps is 5000, which is identical to that of PromptFix.

