# OpenReview forum: "PromptFix: Few-shot Backdoor Removal via Adversarial Prompt Tuning"
_ICLR.cc/2024/Conference — ICLR 2024 Conference Withdrawn Submission_

### Official Review · Reviewer_7Ta8 · 2023-10-31

**Soundness:** 2 fair
**Presentation:** 2 fair
**Contribution:** 3 good
**Rating:** 5
**Confidence:** 4

**Summary:**

This paper proposes to remove the backdoor by adversarial prompt-tuning. , PromptFix keeps the model parameters intact and only utilizes two extra sets of soft tokens that approximate the trigger and counteract it respectively. Another potential property of this method could be its transferability to backdoor attacks (w. new triggers), although this part is not covered by this paper.

**Strengths:**

+ Considering prompt tuning for mitigating the backdoor effect is an interesting idea.

+ Experiments on multiple datasets show the effectiveness of the new defense method in comparison with DBS.

**Weaknesses:**

+ This paper requires careful polishing for a better reading experience. The motivations of several designs are not clearly explained, e.g., bi-level optimization and $L_{CLS}$.

+ The novelty of the technology involved in this work is limited. It is applying prompt-tuning to two-stage backdoor removal.

+ I think this work could be a nice work on improving the robustness against new backdoor attacks, but there is limited discussion on this direction.

+ DBS is considered a baseline, can you explain why it is the only possible baseline?

**Questions:**

**Main Questions:**

The steps for these two steps: num\_round and num\_promppt\_step are ad-hoc to me. Can you explain how to choose these parameters, especially when we are not aware of the attack strategies or hyper-parameters in new application scenarios or new tasks?

The CE threshold loss is ad-hoc. Why it is the best choice? Can you prove it theoretically or empirically?

---

**Minor Issues:**

Page 2: 'to' breach

The meanings of $w_p$ and $w_{CLS}$ should be explained before Section 3.4.

You may shuffle the order of sections 3.4 and 3.5.

Split Acc and ASR in Table 10.

---

### Official Review · Reviewer_Qy5u · 2023-10-31

**Soundness:** 2 fair
**Presentation:** 2 fair
**Contribution:** 3 good
**Rating:** 5
**Confidence:** 4

**Summary:**

This paper introduces PromptFix: a method to defend against backdoor attacks on NLP models. Classic ways to defend against backdoors (where inserting some string into the input consistently produces a specific classification labels) involve trying to identify backdoors, then fine-tuning them away. In contrast, PromptFix optimizes to defend against the _worst-case_ adversarial backdoor, which avoids the lossy backdoor identification step. PromptFix relies on many add-ons to successfully do this, including jointly optimizing for clean accuracy, regularizing, adding a cross entropy loss threshold, and using unlabeled data for regularizations. The authors find that PromptFix has higher clean accuracy than existing defenses, and reduces the attack success rate by more. Moreover, PromptFix outperforms existing methods under domain shift settings and stealthiness settings, which are more realistic.

**Strengths:**

* The motivation for PromptFix are very clear, and the method itself is interesting conceptually; rather than trying to identify backdoors, just optimize over the worst one. This seems like an approach that should scale much better.
* The empirical results are really strong; the authors do a great job evaluating over lots of settings (e.g., different amounts of training data, distribution shift, different backdoor types), and consistently find that PromptFix has better clean accuracy and a lower attack success rate than existing baselines.
* The method presented is a nice application of PGD.

**Weaknesses:**

* The main weakness of the paper is the lack of clarity in writing; as representative examples, it's unclear whether the authors optimize over discrete tokens or token embeddings, and it's unclear what the "few-shot" data is doing.
* The related work section seems incomplete and has misleading references. For example, the authors say AutoPrompt uses "soft-prompts" with tunable embeddings, but AutoPrompt largely uses discrete tokens. On the other hand, prefix tuning (Li and Liang, 2021; 1600 citations) introduces the main method to optimize over the continuous token embeddings, and is not cited.
* There are many additional steps added to the core idea of PromptFix to get it to work, leading to potentially unfair comparisons with baselines.

**Questions:**

* Do you optimize over soft-prompts, or discrete tokens? I'm assuming soft-prompts, but it would be nice to clarify this.
* In equation 1, should you be maximizing over t? And is D actually a distribution, or a few-shot set of samples?

---

### Official Review · Reviewer_H931 · 2023-11-05

**Soundness:** 2 fair
**Presentation:** 2 fair
**Contribution:** 2 fair
**Rating:** 3
**Confidence:** 3

**Summary:**

This paper proposes a new method, PromptFix, to better approximate adversarial triggers and remove the backdoor with minimal training samples. They aim to remove the backdoor with a bi-level optimization problem, learning soft tokens with prompt-tuning. Evaluation is done on Round 6 of the TrojAIIARPA dataset, which consists of binary text classification with backdoored models on the AmazonReview dataset. The performance of PromptFix is compared with DBS, Dynamic Bound-scaling. Performance improvements are shown on the Round 6 dataset in 2-shot and 4-shot settings, both for clean accuracy and ASR.

**Strengths:**

-There is significant value in using lightweight tuning for backdoor removal, as it should be a goal of the community to defend against attacks using less resources and only a few training examples. Better aligning the current methods within the literature for backdoor removal with broader advances in NLP (like lightweight tuning) should allow for a rise in performance in such a domain. In the paper it is shown DBS does not work as well when there are few training samples, which seems much more likely when considering the scope and intentions of the problems. The PromptFix methodology largely builds upon DBS and similar two-stage backdoor removal methods but merges the optimization steps into a bi-level problem to align with the low data setting and utilizes soft tokens to better approximate the trigger, which doesn’t face the same constraints as hard tokens.

**Weaknesses:**

-The experiment part is incomplete and unconvincing. DBS (Shen et al., 2022) is used as the only baseline, and it’s mentioned that DBS assumes access to a benign model for reference which isn’t reasonable for this task. Why choose this task then or at least not include more? They also mention that DBS has many more learnable parameters, which to me explains the worse performance when training on such little data. Overall, it seems that DBS is not the best baseline for this task, and definitely should not be the only one as there are lighter learning methods that have already shown a degree of effectiveness here. Additionally, the experimental results seem too constrained to make larger conclusions about performance. The scope of the evaluation is much smaller than for DBS: DBS evaluates on 1600 models while the authors only use 100, and they evaluate on 3 NLP tasks while the authors only have one (though you evaluate on IMDB out of domain, which is a good start), and they have 7 architectures while the authors are only using DistilBert.

-For the ablation study regarding few-shot settings, performance on PromptFix is tested on 20 examples and then it’s concluded that PromptFix is also applicable when there are more samples. However, since PromptFix is tailored towards few-shot results, I also want to see DBS’ performance on 20 samples, as it may be enough to beat PromptFix.

**Questions:**

As described above, I think more experiments are needed over a wider assortment of datasets, as well as more comprehensive baselines as exhibited in the DBS paper. Also, as mentioned, I am confused why this dataset was selected alone as it doesn’t seem DBS is fully meant for this (though round 6 is in the DBS paper). Clarifying the motivations behind this choice would be helpful.


A few smaller things to improve clarity:
- Define few-shot tuning explicitly as it mainly reminds me of in-context learning, which can be confusing when you mention few-shot prompting too.


- Figures and tables need more descriptive captions; e.g., Figure 1 is not referenced in the paper until page 4, and even that is not explained thoroughly.


- It’s hard to see the improvements in accuracy and ASR as none of the values of the metrics are mentioned at all in the experiment section and the best number is not bolded in the tables.


- More proofreading is needed as some of the paper has grammatical errors or reads awkwardly; e.g., the table should be capitalized to Table when referencing one.